

# Tropospheric ozone radiative forcing uncertainty due to pre-industrial fire and biogenic emissions

Matthew J. Rowlinson[1], Alexandru Rap[1], Douglas S. Hamilton[2], Richard J. Pope[1,3], Stijn Hantson[4,5], Stephen R. Arnold[1], Jed O. Kaplan[6,7,8], Almut Arneth[4], Martyn P. Chipperfield[1,3], Piers M. Forster[9], Lars Nieradzik[10]

[1]Institute for Climate and Atmospheric Science, School of Earth and Environment, University of Leeds, Leeds, LS2 9JT, UK.
[2]Department of Earth and Atmospheric Science, Cornell University, Ithaca 14853 NY, USA.
[3]National Centre for Earth Observation, University of Leeds, Leeds, LS2 9JT, UK.
[4]Atmospheric Environmental Research, Institute of Meteorology and Climate research, Karlsruhe Institute of Technology, 82467 Garmisch-Partenkirchen, Germany.
[5]Geospatial Data Solutions Center, University of California Irvine, California 92697, USA.
[6]ARVE Research SARL, Pully 1009, Switzerland.
[7]Environmental Change Institute, School of Geography and the Environment, University of Oxford, Oxford OX1 3QY, UK.
[8]Max Planck Institute for the Science of Human History, Jena 07745, Germany.
[9]Priestley International Centre for Climate, University of Leeds, LS2 9JT, Leeds, UK.
[10]Institute for Physical Geography and Ecosystem Sciences, Lund University, Lund S-223 62, Sweden.

Corresponding authors: Matthew J. Rowlinson (ee11mr@leeds.ac.uk); Alex Rap (a.rap@leeds.ac.uk)



**Abstract** Tropospheric ozone concentrations are sensitive to natural emissions of precursor compounds. In contrast to existing assumptions, recent evidence indicates that terrestrial vegetation emissions in the pre-industrial were larger than in the present-day. We use a chemical transport model and a radiative transfer model to show that revised inventories of pre-industrial fire and biogenic emissions lead to an increase in simulated pre-industrial ozone concentrations, decreasing the estimated pre-

industrial to present-day tropospheric ozone radiative forcing of up to 34% (0.38 Wm$^{-2}$ to 0.25 Wm$^{-2}$). We find that this change is sensitive to employing biomass burning and biogenic emissions inventories based on matching vegetation patterns, as co-location of emission sources enhances the effect on ozone formation. Our forcing estimates are at the lower end of existing uncertainty range estimates (0.2 – 0.6 Wm$^{-2}$), without accounting for other sources of uncertainty. Thus, future work should focus on reassessing the uncertainty range of tropospheric ozone radiative forcing.




# 1 Introduction

Tropospheric ozone ($O_3$) is a short-lived greenhouse gas formed in the atmosphere through photochemical oxidation of volatile

organic compounds (VOCs) in the presence of nitrogen oxides ($NO_x$). These precursor gases have both natural and anthropogenic sources, and increased anthropogenic emissions are thought to have caused an increase in global tropospheric $O_3$ of 25-50% since 1900 (Lamarque et al., 2010; Gauss et al., 2006; Young et al., 2013). The Intergovernmental Panel on Climate Change (IPCC) current best estimate for tropospheric $O_3$ radiative forcing (RF) over the industrial era is $0.4 \pm 0.2$ $Wm^{-2}$ with a 5%-95% confidence interval , making tropospheric $O_3$ the third most important anthropogenic greenhouse gas

after $CO_2$ and $CH_4$ (Myhre et al., 2013). The present-day (PD) radiative effect (RE) of tropospheric $O_3$ is relatively well constrained (Rap et al., 2015). The large uncertainty range (0.2-0.6 $Wm^{-2}$) is caused by a number of factors such as the radiative transfer scheme employed, the model used to simulate tropospheric $O_3$ and tropopause definition, however it is primarily associated with a poor understanding of pre-industrial (PI) $O_3$ concentrations (Myhre et al., 2013; Stevenson et al., 2013). Although measurements of tropospheric $O_3$ exist as far back as the late 19th century, there are limited reliable quantitative

measurements of tropospheric $O_3$ prior to the 1970s (Volz and Kley, 1988; Cooper et al., 2014). Recently Checa-Garcia et al. (2018) found that differences in PI estimates between Coupled Model Intercomparison Project phase 5 (CMIP5) and CMIP6 cause an 8-12% variation in $O_3$ RF estimates, but did not explicitly assess uncertainty in natural PI emissions. Recent analysis of oxygen isotopes in polar ice cores indicates that tropospheric $O_3$ in the northern hemisphere increased by less than 40% between 1850 and 2005, indicating that $O_3$ RF may be lower than the 0.4 $Wm^{-2}$ estimate (Yeung et al., 2019).


As well as anthropogenic sources, $O_3$ precursor gases such as methane ($CH_4$), carbon monoxide (CO) and $NO_x$ have natural emission sources, e.g., wildfires, wetlands, lightning and biogenic emissions. Wildfires, for example, emit large quantities of CO, $NO_x$, $CH_4$ and non-methane volatile organic compounds (NMVOCs) (van der Werf et al., 2010; Voulgarakis and Field, 2015), which influence the chemical production of $O_3$ (Wild, 2007). Changes in the natural environment therefore influence

the concentration and distribution of tropospheric $O_3$ (Monks et al., 2015; Hollaway et al., 2017). However, the human impact on natural emissions over the industrial era is more uncertain than on anthropogenic emissions (Arneth et al., 2010; Mickley et al., 2001). An accurate representation of PI fire occurrence is required for PI to PD tropospheric $O_3$ RF calculations.

Recent studies suggest that the relationship between humans and fire (Bowman et al., 2009) is more complex than previously

assumed (Doerr and Santín, 2016). The expansion of agriculture and land-cover fragmentation since PI has decreased the abundance and continuity of fuel, inhibiting fire spread (Swetnam et al., 2016; Marlon et al., 2008) and hence total emissions. Furthermore at the global scale, increased population density results in declining fire frequency (Knorr et al., 2014; Andela et al., 2017). Increased agricultural land coupled with active fire suppression and management policies mean that human activity has likely caused total fire emissions to decline since the PI (Marlon et al., 2016; Hamilton et al., 2018; Daniau et al., 2012).

Paleoenvironmental archives of fire activity also reflect a decline of fire over the industrial era in many regions (Swetnam et

al., 2016; Rubino et al., 2016; Marlon et al., 2016). This change in understanding of PI fire emissions has been shown to have a strong influence on aerosol RF: Hamilton et al. (2018) estimated a 35-91% decrease in global mean cloud albedo forcing over the industrial era when using revised PI fire emission inventories.

Emissions of biogenic VOCs (BVOCs), such as isoprene and monoterpenes, from vegetation also affect tropospheric $O_3$ formation. Isoprene contributes to the formation of peroxyacetylnitrate (PAN), which has a lifetime of several months in the upper troposphere (Singh, 1987), allowing long-range transport of reactive nitrogen and enhancing $O_3$ formation in remote regions. PAN formation is also highly dependent on $NO_x$ concentrations, meaning that changes in distribution of emissions as well as the magnitude will impact $O_3$ formation. Previous studies of PI tropospheric $O_3$ have often assumed that PI BVOC

emissions were equivalent or lower than those in PD (Stevenson et al., 2013). In Stevenson et al. (2013), only one model of the ensemble included PI isoprene emissions that were larger than in the PD simulation. However, BVOC emissions are sensitive to climate, $CO_2$ concentrations, vegetation type, and foliage density; each of which has changed since the PI (Hantson et al., 2017; Laothawornkitkul et al., 2009) and needs to be accounted for when calculating $O_3$ RF.

Here our aim is to examine the effect of revised PI fire and BVOC emission inventories on PI-PD tropospheric $O_3$ RF estimates. We use a global chemical transport model (CTM) and a radiative transfer model to investigate the impact of these improved natural PI emission inventories on PI tropospheric $O_3$ and how changes in concentration subsequently alter $O_3$ PI-PD RF. The IPCC 5[th] assessment report moved to the concept of effective radiative forcing (ERF) (Myhre et al., 2013) to more completely capture the expected global energy budget change from a given driver. However, here we employ the more traditional

stratospherically adjusted RF as it can be estimated with more certainty than ERF and previous studies suggest that ERF and RF are likely to be similar for $O_3$ change (Myhre et al., 2013; Shindell et al., 2013). A number of factors not considered here also introduce uncertainty when simulating PI tropospheric $O_3$ concentrations, such as changes to lightning and soil $NO_x$ emissions, $O_3$ deposition and atmospheric transport. The purpose of this study is to quantify the uncertainty associated with natural emissions in the pre-industrial, utilising the revised inventories of fire and biogenic emissions.

## 2 Methods

### 2.1 TOMCAT-GLOMAP

We used the TOMCAT global three-dimensional offline chemical transport model (CTM) (Chipperfield, 2006) coupled to the GLOMAP modal aerosol microphysics scheme (Mann et al., 2010) to simulate tropospheric composition and its response to emissions changes. The model used a 2.8°×2.8° horizontal resolution with 31 vertical levels from the surface to 10 hPa. All

simulations were run with 6-hourly 2008 meteorology from European Centre for Medium-Range Weather Forecasts (ECMWF) ERA-Interim reanalyses with a 1-year spin-up (Dee et al., 2011). The model includes a detailed representation of hydrocarbon chemistry and isoprene oxidation, and has previously been shown to accurately reproduce observed concentrations and





distributions of key tropospheric species such as $O_3$, CO, $NO_x$ and VOCs (Monks et al., 2017; Rowlinson et al., 2019). Biomass burning and biogenic emissions are emitted into the lowest model level.

**2.2 Radiative transfer model**

Tropospheric $O_3$ RFs were calculated using the SOCRATES radiative transfer model (Edwards and Slingo, 1996) with six bands in the shortwave (SW) and nine in the longwave (LW). This version has been used extensively in conjunction with TOMCAT-GLOMAP for calculating $O_3$ radiative effects (Kapadia et al., 2016; Scott et al., 2018; Bekki et al., 2013). We used the fixed dynamical heating approximation (Fels et al., 1980) to account for stratospheric temperature adjustments, i.e. changes 105 in stratospheric heating rate calculated in the model due to the $O_3$ perturbation are applied to the temperature field, with the model run iteratively until stratospheric temperatures reach equilibrium (Forster and Shine, 1997; Rap et al., 2015).

| Simulation | Fire emissions | Biogenic emissions |
|---|---|---|
| PD CMIP6 | GFEDv4 | CCMI |
| PI CMIP6 | CMIP6 | CCMI |
| PI SIMFIRE-BLAZE | SIMFIRE-BLAZE | CCMI |
| PI LMfire | LMfire | CCMI |
| PI CMIP6-BIO | CMIP6 | LPJ-GUESS |
| PI SIMFIRE-BLAZE-BIO | SIMFIRE-BLAZE | LPJ-GUESS |
| PI LMfire-BIO | LMfire | LPJ-GUESS |

**Table 1. Details of TOMCAT-GLOMAP simulations. All simulations are run with present-day meteorology with a one-year spin-**
**up.**

**2.3 Simulations**

In order to investigate the effect of natural PI emissions on PI to PD changes in tropospheric $O_3$ concentrations, we performed one PD and six PI TOMCAT simulations. The PD simulation used the Global Fire Emissions Database (GFED) version 4s 115 (GFED v4s) inventory (as employed in CMIP6) (Randerson et al., 2017; van Marle et al., 2017), biogenic emissions from Chemistry-Climate Model Initiative (CCMI) (Sindelarova et al., 2014) and anthropogenic emissions from the MACCity emissions dataset (From EU projects MACC/CityZEN; Lamarque et al., 2010).


Global mean TOMCAT surface $CH_4$ concentrations are scaled to be 1789 ppb in PD and 722 ppb in the PI (McNorton et al.,
2016; Etheridge et al., 1998; Dlugokencky et al., 2005; Hartmann et al., 2013). In all PI simulations, anthropogenic emissions
are zero except biofuel emissions taken from AeroCom for the year 1750 (Dentener et al., 2006). The first set of three
simulations, CMIP6, SIMFIRE-BLAZE and LMfire, investigated the impact of fire emissions only by keeping PI BVOC
emissions (i.e. isoprene and monoterpenes) at PD values (Table 1). The second set of three simulations, CMIP6-BIO,
SIMFIRE-BLAZE-BIO and LMfire-BIO, investigated the additional impact of PI biogenic emissions, by combining each PI
fire emission inventory with an estimate of PI BVOC emissions from the LPJ-GUESS model (Table 1).

## 2.4 Fire emission inventories

Following Hamilton et al. (2018), we used three PI inventories to investigate the sensitivity of tropospheric $O_3$ RF to PI fire
uncertainty. The CMIP6 PI inventory is treated as a control, as this has been widely used in previous studies and was developed
from a set of global fire models, with SIMFIRE-BLAZE and LMfire providing PI perturbation scenarios from this baseline.

### 2.4.1 CMIP6

CMIP6 provides monthly mean emissions of CO, $NO_x$, $CH_4$ and VOCs from fires. In the PD, CMIP6 emissions are derived
from satellite estimates of global burden area and active fire detections (Giglio et al., 2013; Randerson et al., 2012). In the
absence of satellite data, PI CMIP6 fire emissions are generated by merging PD satellite observations with fire proxy records,
visibility records and analysis from six fire models (van Marle et al., 2017). The mean of 1750-1770 emissions is used in this
study to represent PI emissions. Biomass burning emissions from deforestation and peat fires are assumed to be reduced in the
PI, while agricultural fires are kept fairly constant with PD due to a lack of information (van Marle et al., 2017).

### 2.4.2 Pre-industrial SIMFIRE-BLAZE

The SIMFIRE-BLAZE PI fire emission inventory was developed using the LPJ-GUESS-SIMFIRE-BLAZE model. The PI
emissions here are the mean for the period 1750-1770 (Hamilton et al., 2018). The LPJ-GUESS dynamic vegetation model
predicts ecosystem properties for given climate variables (Smith et al., 2014), which, combined with the HYDE 3.1 dataset of
human land-use change, allows simulation of global PI land cover (Klein Goldewijk et al., 2011). The SIMple fire model
(SIMFIRE) calculates total burned area (Knorr et al., 2014) with total fire carbon-flux calculated from BLAZE (BLAZe
induced biosphere-atmosphere flux Estimator) (Rabin et al., 2017). Akagi et al. (2011) emissions factors were used with
separate treatment of herbaceous and non-herbaceous, tropical and extratropical vegetation to produce emission inventories.
Agricultural fire emissions are not included in SIMFIRE-BLAZE. Total PI fire emissions of gas species in the SIMFIRE-
BLAZE inventory are 28% larger than in the PI CMIP6 inventory.





### 2.4.3 Pre-industrial LPJ-LMfire

The LPJ-LMfire model calculates dry matter consumed by fire and simulates natural wildfire ignition from lightning (Murray
et al., 2014; Pfeiffer et al., 2013). Land use is prescribed for the year 1770 using the KK10 scenario from Kaplan et al. (2011).
Akagi et al. (2011) emissions factors were again used to calculate the gas-phase fire emissions from dry biomass burned in
each grid cell. Burned area is calculated based on fuel availability. LMfire includes emissions from managed agricultural
burning, with 50% of the litter on 20% of used croplands burden annually. Also included are emissions from post-harvest
agricultural burning, with 10% of harvested agricultural crop material assumed to be burned each year. Total PI fire emissions
in LMfire are approximately double the SIMFIRE-BLAZE inventory and four times larger than CMIP6 emissions.

### 2.5 Assessment of PI fire emissions

Despite being significantly larger than CMIP6 and SIMFIRE-BLAZE (Fig. 1), emissions from LMfire have been shown to be
within the quantifiable uncertainty of fire emissions (Lee et al., 2013). Furthermore, LMfire compares more favourably than
the CMIP6 and SIMFIRE-BLAZE inventories with Northern Hemisphere (NH) ice core records in Greenland and Wyoming
(Chellman et al., 2017; Hamilton et al., 2018). In addition to the examination of paleoenvironmental archives with PI fire
emissions datasets by Hamilton et al. (2018), we compared simulated annual mean surface PI CO concentrations in Antarctica
for each fire emissions inventory using the Southern Hemisphere (SH) ice core CO record from Wang et al. (2010). Simulated
Antarctic CO concentrations using PI CMIP6 emissions are 37 ppb, substantially lower than the Wang et al. (2010) 1750 value
of $45 \pm 5$ ppb. This CMIP6 value is closer to the 650-year minimum that occurred in the mid-17th century (38 ppb). When
using SIMFIRE-BLAZE and LMfire emissions, Antarctic CO concentrations for 1750 are estimated at 48 ppb and 61 ppb,
respectively. The overestimation when using LMfire suggest that SH CO emissions may be high for 1750; however, they are
comparable to the peak CO concentration measured in the late 1800s ($55 \pm 5$ ppb) when fire emissions also peaked (van der
Werf et al., 2013). As 1850 is also sometimes used as the PI baseline year when calculating RF, we suggest LMfire provides
a realistic upper bound to possible PI fire emissions.


The combined evaluation of these inventories in Hamilton et al. (2018) and here indicates that although the revised PI fire
inventories differ considerably from each other and are larger than CMIP6, they are closer to proxy records than CMIP6
estimates and therefore their respective impacts on tropospheric $O_3$ RF need to be considered.

### 2.6 Biogenic emission inventories

### 2.6.1 Present-day CCMI

The PD control biogenic emissions were provided from the CCMI inventory. CCMI mean annual BVOC emissions,
comprising isoprene and monoterpenes, are derived using the Model of Emissions of Gases and Aerosols from Nature
(MEGAN) model (Guenther et al., 2012) under the MACC project (Sindelarova et al., 2014). The CCMI inventory estimates



global BVOC emissions at 623 Tg/yr, in reasonable agreement with surface flux measurements and other modelling studies
(Sindelarova et al., 2014; Arneth et al., 2008; Rap et al., 2018).

### 2.6.2 LPJ-GUESS

Alternative biogenic emissions were produced using the LPJ-GUESS dynamic vegetation model simulating isoprene and
monoterpenes (Arneth et al., 2007; Schurgers et al., 2009). Total PD emissions and distribution in the LPJ-GUESS inventory
(i.e. 607 Tg/yr) are similar to the PD CCMI inventory (Fig. 2). For the PI emissions, the LPJ-GUESS biogenic emissions
inventory is based on the mean for the period 1750-1770, estimated to be 836 Tg/yr. There are large spatial differences between
the PI LPJ-GUESS and PD CCMI inventories, with significantly higher emissions in South America and Central Africa, and
lower emissions in South-East Asia in the PI LPJ-GUESS inventory (Fig. 2).

## 3 Results and discussion

### 3.1 Pre-industrial emission inventories

Figure 1a-d shows annual latitudinal fire emissions of CO, $NO_x$, $CH_4$ and VOCs for the CMIP6, SIMFIRE-BLAZE and LMfire
PI inventories, compared to the PD CMIP6 inventory. We also compare BVOC emissions (i.e. isoprene and all monoterpenes)
from the LPJ-GUESS inventory with the PD CCMI inventory. In the PI CMIP6 simulation, global CO emissions have
increased by a factor of 2.5 between PI and PD from 381 Tg/yr to 970 Tg/yr. The main driver of this increase is industrial
emissions, particularly in the NH mid-latitudes. There is large variation in simulated CO emissions between the three PI fire
inventories: 644 Tg/yr in SIMFIRE-BLAZE (69% larger than CMIP6) and 1152 Tg/yr in LMfire (200% larger). Estimates of
CO emissions using LMfire results in total global emissions which are larger than the PD estimate, which also includes
anthropogenic sources. The larger PI biomass burning emissions in LMfire are a result of a number of factors not present in
the other PI inventories such as the inclusion of high-latitude fire occurrence, agricultural fire emissions and differing emission
factors (Hamilton et al., 2018).

Global $NO_x$ emissions also vary considerably between PI inventories, with values in the SIMFIRE-BLAZE inventory
increasing 13% compared to the CMIP6 inventory (36 Tg/yr compared to 32 Tg/yr). This difference is largely due to increased
emission in NH mid-latitudes within SIMFIRE-BLAZE. $NO_x$ emissions in LMfire are 112% larger than the CMIP6 total (68
Tg/yr), with the most significant increases in the extra-tropics.

As $CH_4$ emissions from fires are significantly smaller than CO emissions (Voulgarakis and Field, 2015) increased PI fire
estimates do not substantially alter total $CH_4$ emission. $CH_4$ emissions in SIMFIRE-BLAZE and LMfire are similar in amount
and distribution, 15% and 9% lower than CMIP6, respectively. There is an increase in SH $CH_4$ emissions in both SIMFIRE-
BLAZE and LMfire compared to CMIP6 but a decrease in the NH and SH mid-latitudes. Total PI $CH_4$ emissions are greatest



in CMIP6 at 241 Tg/yr, approximately 43% of PD emissions. Due to the scaling of global mean surface $CH_4$ concentrations in TOMCAT-GLOMAP, the effect of changes in amount of $CH_4$ emitted will be very small, however the change in distribution may impact the formation and loss rates of tropospheric $O_3$.

In terms of fire-emitted VOC species, their size and distribution of emissions are fairly consistent between PD and PI inventories. PI CMIP6 are 87% of PD CMIP6 values, with PI SIMFIRE-BLAZE at 97% (303 Tg/yr). Total global VOC emissions are largest in LMfire at 349 Tg/yr, 29% larger than PI CMIP6 (271 Tg/yr) and 13% larger than PD CMIP6 (310 Tg/yr). The distribution of global VOC emissions is relatively uniform across all inventories, however, individual species do have larger variability between inventories. Formaldehyde and acetylene for example have substantially increased SH emissions in SIMFIRE-BLAZE and LMfire.





**Figure 1: Annual latitudinal mean fire emissions (in Tg/yr) of (a) CO, (b) NO$_x$, (c) CH$_4$ and (d) VOCs and annual zonal mean BVOC emissions (e), for PD (solid black line), PI CMIP6 (dashed green), PI SIMFIRE-BLAZE (dotted orange), PI LMfire (dashed purple), PD LPJ-GUESS (dashed dark green) and PI LPJ-GUESS (dotted light green).**






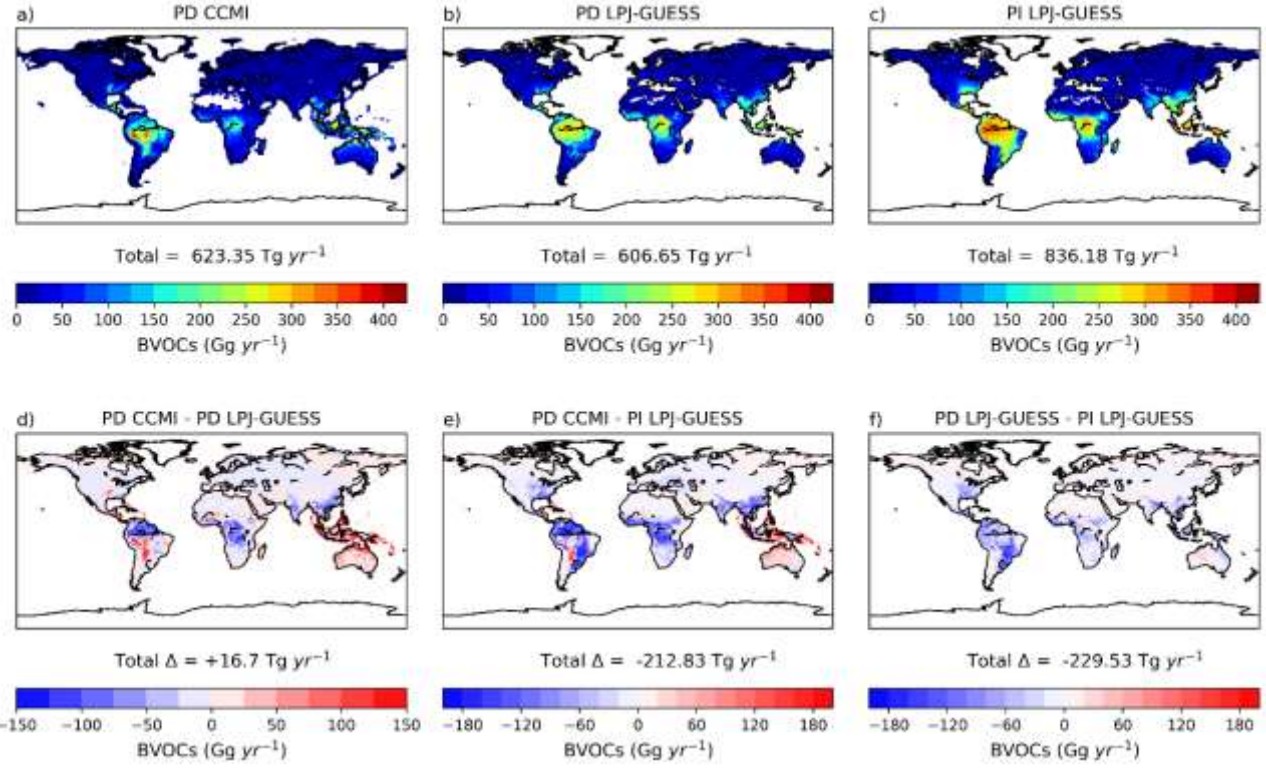

**Figure 2: Annual BVOC (isoprene + monoterpenes) emissions in the two present-day biogenic emissions inventories (CCMI and LPJ-GUESS) and the preindustrial LPJ-GUESS inventory. Top panels (a-c) show total emissions per year, while lower panels (d-f) show differences between the three inventories. Total annual emissions and difference in annual emissions are also shown.**


The BVOC emissions in the two PD inventories (CCMI and LPJ-GUESS) are similar (Fig. 1e), although a small positive NH gradient exists in PD LPG-GUESS compared to PD CCMI. Total BVOC emissions are 16.7 Tg larger in the PD CCMI inventory than PD LPJ-GUESS (Fig. 2). However, the PI LPJ-GUESS BVOC estimate (836 Tg/yr) is 37% larger than its PD equivalent and 34% larger than PD CCMI, although with a similar spatial distribution (Fig. 2). The largest difference is in

South American emissions, where PI LPJ-GUESS emissions are up to 120 Tg larger than PD. The reduction of BVOC emissions between PI and PD is due to a combination of crop expansion, land cover changes and $CO_2$ inhibition (Hantson et al., 2017). Our results are consistent with previous studies reporting between ~25% (Pacifico et al., 2012; Hollaway et al., 2017; Lathière et al., 2010) and ~35% (Unger, 2014) larger PI values than PD.







**Figure 3: Total monthly fire emissions (in Tg/month) of (a) CO, (b) NOₓ, (c) CH₄ and (d) VOCs and total monthly BVOC emissions**
**(e), for PD (solid black line), PI CMIP6 (dashed green), PI SIMFIRE-BLAZE (dotted orange), PI LMfire (dashed purple), PD LPJ-GUESS (dashed dark green) and PI LPJ-GUESS (dotted light green).**




The seasonality of the fire emissions in the PD and PI inventories used here is demonstrated in Fig. 3. CMIP6 PI and PD emissions have an extremely similar seasonal cycle for all species, with maximum emissions of precursors between July and
September. This is expected as the PI CMIP6 emissions are based on GFED4s climatology and monthly patterns were assumed not to have changed over time (van Marle et al., 2017). This also reflects the PD seasonal cycle of tropospheric $O_3$ in the SH. where anthropogenic contributions are lower than the NH, with maximum concentrations found in September/October (Cooper et al., 2014). The seasonal cycle of CO emissions (Fig. 3a) varies substantially across the 3 PI inventories, with LMfire estimating peak emissions in May-June as opposed to July-August in CMIP6 and SIMFIRE-BLAZE. This may be a result of
increased emissions from SH Africa and Central America, where large fire events are common in late-spring. The inclusion of high-latitude fire occurrence and agricultural burning in LMfire may also play a role, as these contribute to fire emissions in the boreal spring season (Hamilton et al., 2018). The SIMFIRE-BLAZE CO emissions exhibit a similar but more pronounced seasonal cycle to that in CMIP6, with peak emissions in August. Similarly, $NO_x$ and VOC emissions peak earlier in the year in the LMfire inventory relative to SIMFIRE-BLAZE and CMIP6, again with a larger peak in August in SIMFIRE-BLAZE.
Monthly $CH_4$ emissions are broadly consistent across all inventories, with peak emissions in July or August and lower emissions over the NH winter. The seasonality of BVOCs emissions is also consistent across all PI inventories and PD CMIP6, with a peak in July-August. Isoprene emissions are heavily dependent on temperature and photosynthetic active radiation (Malik et al., 2018), therefore reach a maximum in NH summer when there parameters at optimum for vegetation emissions.

Figure 3 indicates similar controls over the modelled seasonality of PI fire occurrence in both PI CMIP6 and PI SIMFIRE-BLAZE, although an increase in estimated fire extent in SIMFIRE-BLAZE resulting in a more pronounced seasonal cycle. LMfire on the other hand estimates a shift in the seasonality of global fire emissions, with peak fire emission occurring earlier than other inventories, as well as a broader peak period of emissions. The change in seasonality of precursors will affect the formation and transport of simulated tropospheric $O_3$ concentrations, as atmospheric chemistry and circulation also have strong
seasonal cycles. However, the broadly similar pattern of maximum emissions in the NH summer and a minimum in NH winter, coinciding with similar climatic conditions, means that the substantial difference in volume of precursor emissions across the PI inventories is likely to be more significant than seasonal changes.





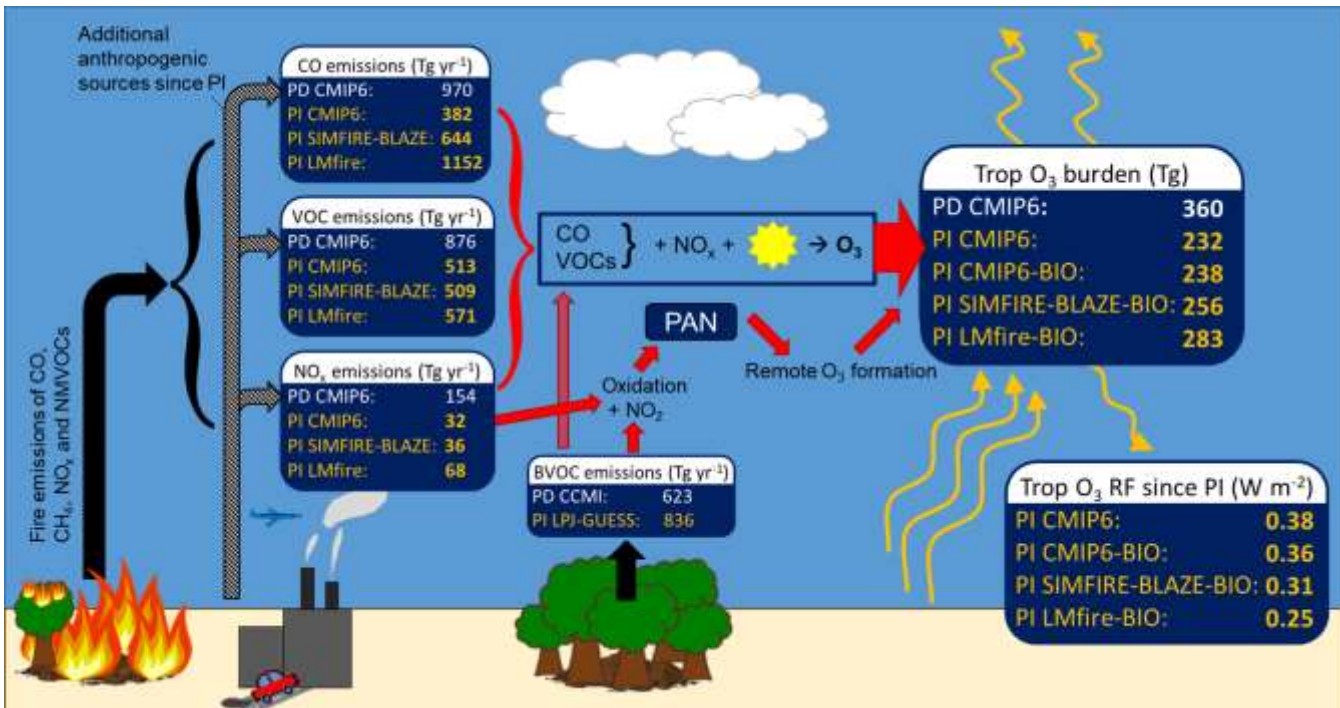

**Figure 4: Summary schematic showing tropospheric O₃ precursor emissions from fire, biogenic and anthropogenic sources, the**
**processes of photochemical O₃ formation, the tropospheric O₃ burden and the PI-PD RF. The magnitude of CO, NOₓ, VOC and**
**BVOC precursor emissions used in this study is shown for the PD (white text) and each PI inventory (yellow text). The resulting**
**calculated tropospheric O₃ burden and RF when using each emission inventory are also shown.**

### 3.2 Pre-industrial fire emissions effect on O₃

Annual emissions of O₃ precursors, their simulated annual mean PI burdens and their contribution to the formation of
tropospheric O₃ are shown in Fig. 4 and Table 2. The largest difference between simulations is in the simulated global
tropospheric CO burden which varies by up to 100 Tg depending on the PI fire emission inventory employed: 195 Tg in the
PI CMIP6 simulation, 232 Tg in PI SIMFIRE-BLAZE (18% higher than CMIP6) and 295 Tg in PI LMfire (50% higher) (Table
2).

The difference in global NOₓ burden between PI simulations is smaller, with increases of 4% and 18% in PI SIMFIRE-BLAZE
and PI LMfire respectively, relative to PI CMIP6. The annual mean NH/SH ratio of tropospheric NOₓ burden in PI simulations
is 1.09, 1.12 and 1.18 for CMIP6, SIMFIRE-BLAZE and LMfire, respectively. Simulated airmass-weighted global mean
concentrations of tropospheric OH, which plays a key role in tropospheric O₃ formation, are 1.06, 1.06 and $1.11 \times 10^6$ molecules
cm⁻³ in CMIP6, SIMFIRE-BLAZE and LMfire, respectively. These values all fall within one standard deviation of the
Atmospheric Chemistry and Climate Model Intercomparison Project (ACCMIP) multi-model mean of $1.13 \pm 0.17$ (Naik et al.,
2013). PI OH concentrations are lower than PD simulated values ($1.12 \times 10^6$ molecules cm⁻³), due to the higher concentrations
of OH precursors NOₓ and O₃ in PD outcompeting the effect of increased CH₄ and CO concentrations which deplete OH (Naik



et al., 2013). The NH/SH OH ratio is $1.25 \pm 0.02$ in the PI simulations compared to 1.41 in the PD CMIP6 simulation, slightly larger than the corresponding ACCMIP multi-model mean values ($1.13 \pm 0.09$ and $1.28 \pm 10$, respectively) but within the inter-
model range and reflecting the expected PI to PD increase (Naik et al., 2013).

Changes to the atmospheric concentration and distribution of $O_3$ precursor species lead to changes in the tropospheric $O_3$ burden. The PI CMIP6 simulation produced the lowest tropospheric $O_3$ burden at 232 Tg, slightly below the ACCMIP multi-model mean of 239 Tg for 1850 (Young et al., 2013). In PI SIMFIRE-BLAZE the burden is 242 Tg (4% higher than CMIP6)
while in LMfire it is 273 Tg (18% higher), slightly outside the range of estimates of 1850 tropospheric $O_3$ burden in ACCMIP models (192 Tg to 272 Tg) (Young et al., 2013). The burdens simulated here represent a PI to PD tropospheric $O_3$ burden change of 55%, 49% and 32% for CMIP6, SIMFIRE-BLAZE and LMfire, respectively. We note that in these simulations the PI LMfire is the only inventory leading to a simulated PI to PD global burden change of less than 40%, a value consistent with that recently indicated by isotope measurements in ice cores (Yeung et al., 2019). The differences between CMIP6 and
SIMFIRE-BLAZE are primarily related to increases in tropospheric $O_3$ within the Amazon region (Fig. 5a). The change in tropospheric $O_3$ vertical profile in the PI SIMFIRE-BLAZE simulation compared to PI CMIP6 (Fig. 5c) shows increased annual mean concentrations throughout the troposphere, driven by changes at 30ºS and 50ºN. Changes between LMfire and CMIP6 simulated tropospheric $O_3$ profiles are larger, with increased $O_3$ at all latitudes. Compared to PI CMIP6, there is a mean global increase in $O_3$ column of 3.7 DU when using LMfire and 1.0 DU when using SIMFIRE-BLAZE. The largest changes
occur over Central Asia, Australia and South America where tropospheric column $O_3$ can be as much as 9.0 DU higher in the PI LMfire simulation than the PI CMIP6 simulation (Fig. 5b). This is reflected in the changes to the vertical $O_3$ profile, with the largest increases in the subtropics. The difference between LMfire and CMIP6 simulations is greatest between 600 and 800 hPa in the SH, and is roughly constant with respect to changes in altitude over the northern subtropics. The only regions where tropospheric $O_3$ is higher in the CMIP6 simulation are Central Africa and Indonesia, likely due to the PI CMIP6
emissions being anchored to PD fire observations and thus transferring these patterns to the PI (van Marle et al., 2017).

The effect of different fire emission inventories on $O_3$ burden is significantly smaller than the impact on CO concentrations (Table 2), as fire emissions are one of several sources of $O_3$ variability (Lelieveld and Dentener, 2000). $O_3$ production is reliant on a number of precursors which do not respond uniformly to the different estimates of fire occurrence in the inventories used
here. The relatively minor response of $NO_x$ concentrations across the three PI emissions estimates (Table 2), and the prevailing $NO_x$-limited state across rural environments in PD (Duncan et al., 2010), suggests that increases in CO and VOCs have only a small impact on $O_3$ production because of $NO_x$ availability limitations. Moreover, Stevenson et al. (2013) attributed the majority of the PI to PD shift in tropospheric $O_3$ to $NO_x$ and $CH_4$ changes, with a relatively small contribution from CO and NMVOCs despite increasing emissions of both. However, the simulated changes still represent significant shifts in the
abundance and distribution of tropospheric $O_3$ in the PI atmosphere.





|  | CO burden (Tg) | NOx burden (Tg) | Mean tropospheric OH (x10^6 mol cm^-3) | O3 burden (Tg) | Tropospheric column O3 (DU) | O3 RF 1750-2010 (Wm^-2) |
|---|---|---|---|---|---|---|
| PD CMIP6 | 342.6 | 73.2 | 1.12 | 359.9 | 31.0 | - |
|  |  |  |  |  |  |  |
| PI CMIP6 | 195.5 | 44.8 | 1.06 | 231.7 | 19.9 | 0.38 |
| PI SIMFIRE-BLAZE | 231.5 | 46.7 | 1.06 | 241.6 | 20.9 | 0.35 |
| PI LMfire | 295.0 | 52.8 | 1.11 | 272.7 | 23.6 | 0.27 |
|  |  |  |  |  |  |  |
| PI CMIP6-BIO | 238.7 | 44.3 | 1.00 | 237.8 | 20.2 | 0.36 |
| PI SIMFIRE-BLAZE-BIO | 283.4 | 46.7 | 1.00 | 256.0 | 22.1 | 0.31 |
| PI LMfire-BIO | 337.1 | 53.4 | 1.08 | 282.8 | 24.4 | 0.25 |

**Table 2:** Annual mean global tropospheric burdens of CO, NOx and O3, mean tropospheric OH concentration, tropospheric column O3 in Dobson units (DU) and radiative forcing of tropospheric O3 1750-2010 for present-day simulation and each PI fire and biogenic emission inventory.



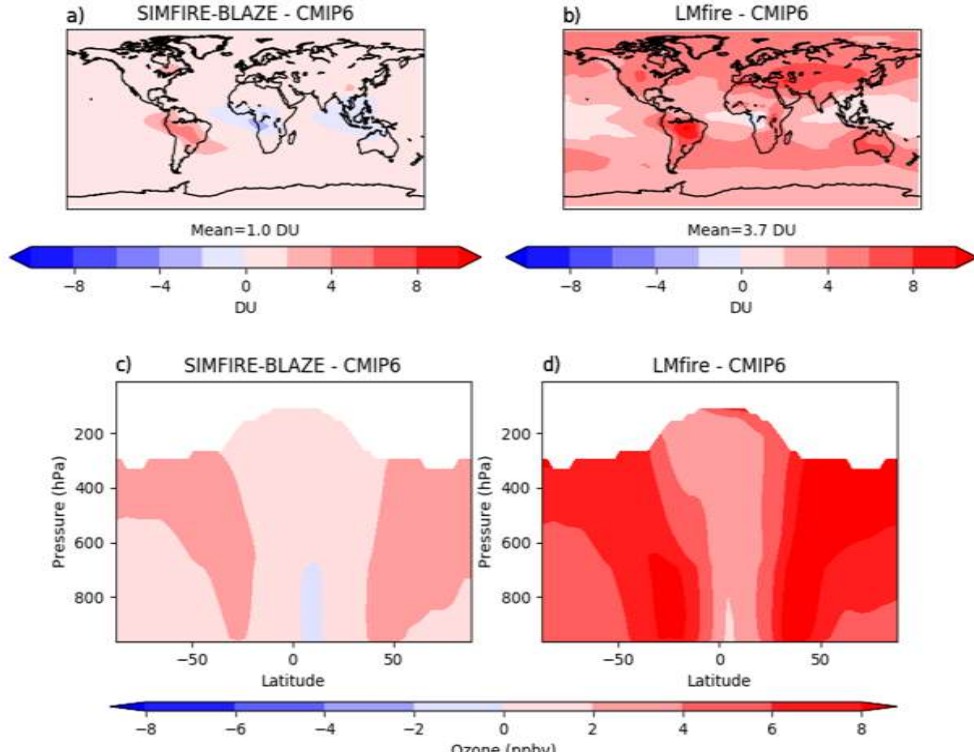

**Figure 5: Difference in simulated PI O₃ between revised inventories SIMFIRE-BLAZE and LMfire and the CMIP6 control. Top panels (a, b) compare differences in tropospheric column O₃ in DU, lower panels (c, d) show differences in zonal mean vertical O₃ in ppbv.**

### 3.3 Pre-industrial BVOC emissions effect on O₃

We repeated the three PI simulations, replacing the PD biogenic emissions with the PI LPJ-GUESS inventory. In general, the
inclusion of PI BVOC emissions increases PI O₃ concentrations, due to an increased VOC source and hence PAN formation
(Fig. 4). For CMIP6 fire emissions, the inclusion of PI BVOCs increases the CO burden by 22% and tropospheric O₃ burden
by 3%, while mean tropospheric OH concentration decreases by 6%. The decrease in OH is the most likely reason for the
simulated increases in CO and O₃. The inclusion of PI BVOCs in the LMfire fire emission simulation causes a 3% decrease in
tropospheric OH, and increases in tropospheric CO and O₃ of 14% and 4%, respectively.

For SIMFIRE-BLAZE, the inclusion of PI BVOCs decreases OH by 6% and increases CO and O₃ by 22% and 6%, respectively.
In all simulations the inclusion of PI BVOCs has only a small effect on the $NO_x$ burden (~1%). The effect on tropospheric O₃
of including PI BVOCs is notably larger in the simulation using SIMFIRE-BLAZE fire emissions compared to CMIP6 or
LMfire. The SIMFIRE-BLAZE simulation combines fire and biogenic emissions produced using the same land-use model,
with consistent vegetation distributions. The co-location of isoprene and $NO_x$ emissions promotes PAN formation, enabling
long-range transport of $NO_x$ and enhancing O₃ production (Hollaway et al., 2017). This synergistic effect has been found to
amplify the effect of biogenic emissions on tropospheric O₃ production (Bossioli et al., 2012). Therefore, if PI biogenic
emissions inventories were specifically produced for each fire inventory, the corresponding impact on O₃ would likely be
larger than presented here. With the inclusion of PI BVOC emissions, both the SIMFIRE-BLAZE and LMfire simulations



result in a PI to PD tropospheric $O_3$ burden change of 40% or less, in line with estimates from ice core observations (Yeung et al., 2019).

### 3.4 Effect on ozone radiative forcing

The estimated tropospheric $O_3$ RF, based on the CMIP6 PI and PD control simulations, is 0.38 Wm$^{-2}$ (Fig. 4 and Table 2), comparing well with the IPCC AR5 estimate of 0.4 ± 0.2 Wm$^{-2}$ (Myhre et al., 2013) and the ACCMIP multi-model mean of

0.41 ± 0.12 Wm$^{-2}$ (Stevenson et al., 2013; Myhre et al., 2013). When PI SIMFIRE-BLAZE and PI LMfire emissions are used instead of PI CMIP6 fire emissions, larger PI tropospheric $O_3$ concentrations lead to 8% (to 0.35 Wm$^{-2}$) and 29% (to 0.27 Wm$^{-2}$) decreases in $O_3$ RF, respectively. When the PI BVOC emission inventory is used in conjunction with each PI fires emission inventory, $O_3$ RF is further reduced compared to the control by 5% (to 0.36 Wm$^{-2}$), 18% (to 0.31 Wm$^{-2}$) and 34% (to 0.25 Wm$^{-2}$), for CMIP6, SIMFIRE-BLAZE and LMfire, respectively (Fig. 4). While these reductions in $O_3$ RF are still within the IPCC

uncertainty range, they are caused entirely by uncertainty in PI precursor emissions from wildfires and vegetation. Other key sources of uncertainty (e.g. inter-model spread, use of different radiative transfer schemes) are not accounted for here and would therefore alter estimates further, potentially outside the current 5%-95% confidence range. The most important region for changes to the RF of $O_3$ is the upper troposphere at subtropical latitudes (Fig. 5d), where there are substantially higher $O_3$ concentrations in the LMfire simulation. $O_3$ changes in this region are up to 10 times more efficient at altering the radiative

flux than in other regions (Rap et al., 2015). However, the lack of a vertical distribution to fire emissions in TOMCAT affects the simulated changes to the $O_3$ vertical profile. Previous studies which introduced an injection height scheme found small increases in $O_3$ production downwind of emission sources (Jian and Fu, 2014), although the change to total $O_3$ and precursors is relatively small (Bossioli et al., 2012; Zhu et al., 2018).

### 4 Conclusions

Revised inventories of PI fire and biogenic emissions substantially decrease estimates of PI to PD tropospheric $O_3$ RF. When using PI LMfire fire emissions, which represent a plausible upper emissions limit, $O_3$ RF is reduced to 0.27 Wm$^{-2}$, 29% smaller than the CMIP6 simulation. Large increases in estimated PI fire occurrence drives increases in PI $O_3$ concentrations (3.7 DU global mean tropospheric column $O_3$ increase for LMfire inventory) through larger emissions of CO, $NO_x$ and VOCs. PI CO increases by up to 51% depending on the PI inventory, but the effect on $O_3$ production is limited by the relatively small increase

in $NO_x$ (~4%). Using PI biogenic emissions, rather than assuming PD values, further increases simulated PI tropospheric $O_3$, though the magnitude of this depends on the fire inventory. When accounting for revised emissions from fire and biogenic sources, both the LMfire and SIMFIRE-BLAZE inventories simulated a PI to PD change in tropospheric $O_3$ burden of approximately 40% or less, in good agreement with estimates from Yeung et al. (2019). Consequently, we find that the estimate of $O_3$ RF since PI decreases by up to 34% (to 0.25 Wm$^{-2}$) when considering the uncertainty in PI emissions of both fires and

BVOCs.

The impact on tropospheric $O_3$ from uncertainty in PI natural emissions suggests that previous estimates of $O_3$ RF over the industrial era are likely too large. Our revised tropospheric $O_3$ RF estimates are at the lower end of the existing uncertainty range, without yet taking into account other sources of uncertainty. We therefore argue that the impact of uncertainty in PI

natural emissions should be further investigated using more models, in order to reassess the current best-estimate and uncertainty range of $O_3$ RF.



**Acknowledgements**

MJR is funded by a NERC SPHERES DTP (NE/L002574/1) studentship. This work used the UK ARCHER (http://www.archer.ac.uk) and Leeds ARC3 high performance computing facilities. RP is funded by the UK National Centre

for Earth Observation (NCEO). SH and AA acknowledge support from EU FP7 projects BACCHUS (grant agreement No. 603445) and LUC4C (grant agreement No. 603542). DSH is funded by the Atkinson Center for a Sustainable Future at Cornell University. PF supported by NERC grant NE/N006038/1 (SMURPHS) and EU Horizon 2020 program grant agreement number 820829 (CONSTRAIN). Datasets available via Open Science Framework (https://osf.io/98c2n/).

**Author contribution**

MJR, AR, DSH and RJP conceptualised the study and planned the model experiments. Emission inventories were produced by SH, JOK, AA and LN, and processed for use in TOMCAT-GLOMAP by RJP and DSH. All model runs and analysis was performed by MJR with guidance from AR, RJP and SRA. The manuscript was written by MJR with comments and advice from all co-authors.






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
