# Peer review of "Tropospheric ozone radiative forcing uncertainty due to preindustrial fire and biogenic emissions"

_Atmospheric Chemistry and Physics, 2019_

## Referee Comment (RC1) · Anonymous Referee #1 · 11 Feb 2020

In this study, the authors use a chemistry transport and different inventories of preindustrial fire and biogenic emissions to argue that the uncertainty range of ozone radiative forcing has been overestimated in past multi-model studies and assessments. The paper is the ozone counterpart to Hamilton et al. (2018), which made a similar point about biomass-burning aerosols.

The paper is very well written and structured in a straightforward way. The changes in simulated tropospheric ozone are well understood from differences in precursor emissions, so the question is whether the alternative sets of preindustrial emissions are a good guide to the overall uncertainty. This is where my concerns are, as detailed be-

low. Addressing my comments may involve new simulations, so may represent major revisions.

**1 Main comments**

- My main concern with the study is that the PD/PI pairs used to estimate radiative forcing are not consistent. There is only one PD simulation, using the CMIP6 inventory. But shouldn't the SIMFIRE-BLAZE PI simulation be coupled with a SIMFIRE-BLAZE PD simulation? Shouldn't the LMfire PI simulation be coupled with an LMfire PD simulation? If the PD simulations differ from CMIP6 in the same way as the PI simulations, then the impact on radiative forcing would be small. I acknowledge that fire models (including those used to provide the CMIP6 inventory) are typically overfitted to present-day observations, so their PD simulations should share common patterns, but at least the PD and PI distributions would always be consistent in terms of the internal physics of the fire emissions.

- In a related concern, I note that section 2.6 implies that CCMI is a reasonable biogenic emission inventory for present-day because it compares well to flux measurements and other models. Then LPJ-GUESS is said to be similar to CCMI for present-day, implying it is also a reasonable inventory. Those are weak arguments, but there is at least an attempt at looking at performance of inventories. In contrast, section 2.4 on fire emission inventories does not discuss present-day performance. This is a problem because if SIMFIRE-BLAZE and/or LPJ-LMfire happen to be biased in an era where they can be constrained by observations, then the authors overstate the case for preindustrial emission uncertainty.

**2 Other comments**

- Line 158: "within the quantifiable uncertainty of fire emissions (Lee et al., 2013)". What do the authors mean here? For present-day or preindustrial? And is Lee et al. the correct reference? That paper does not mention LMFire at all.

- Figure 1a: LMfire has large CO emissions between 25 and 50S. What is burning there? Australia? Argentina?

- Figures 2a,b,c: What are those black lines in South America and Africa? In the difference maps, they seem to correspond to a brutal change in emissions, with differences between datasets switching sign suddenly.

---

## Referee Comment (RC2) · Anonymous Referee #2 · 2 Mar 2020

GENERAL POINTS

This is an interesting study – and makes an important point: pre-industrial emissions from fires and biogenic sources are a major source of uncertainty for ozone radiative forcing. As explained below, it could benefit from some clarifications. In particular, why are these new estimates of PI emissions better than those used by CMIP6? Some details of the modelling need to be clarified – I was baffled by the discussion of CH4 emissions for simulations where I thought CH4 concentrations were prescribed. If the points below can be cleared up, then I am happy to recommend this paper should be accepted for publication in ACP.

[Figure]

SPECIFIC POINTS

L25 of up to -> by up to

L56 "human impact on. . . anthropogenic emissions. . ." Reword. I think we can be fairly sure there is a human impact on anthropogenic emissions. . .

L99 State thickness (metres or hPa) of the lowest model level.

L119 Do the prescribed surface CH4 concentrations have spatial variation, or just a constant value everywhere? Given later comments about CH4 emissions, please clarify further how CH4 is handled by the model.

L146 "Total PI fire emissions. . . SIMFIRE-BLAZE. . . are 28% larger than. . . PI CMIP6". It would be instructive to know PD fire emissions predicted by the SIMFIRE-BLAZE model. Can the model reproduce the present-day GFED distribution, or something similar? It may be that the higher PI values indicate a bias in this model towards higher values. It is hard to know how to verify or evaluate the PI fire emissions without some measure of the model's abilities – and presumably evaluation for present-day is the best evaluation possible. If this is not the case, then at least some discussion of how much faith we should have in these PI values is required.

L155 Similarly for the LPJ-LMfire model.

Perhaps the key question here is whether the fire models used here are better than the fire models used in the CMIP6 base case. Are they clearly better, or are they just different? My non-expert reading of this is that they are just different. Please do try to convince me they are better.

Figure 1, and all the figures, are of a poor resolution. I can just about make out the necessary details, but these need to be improved for the final version.

In Figure 1c, the PD CMIP6 CH4 emissions from fire total 566.6 Tg. This sounds suspiciously high – isn't that more like the value for the total PD CH4 emission flux?

[Figure]

L192 delete PI

L193 "The main driver of this increase [in fire emissions] is industrial emissions. . ." This must be wrong?

L210 I don't understand why CH4 emissions are presented and discussed; surely if CH4 concentrations are prescribed, the CH4 emissions are irrelevant and redundant? Am I missing something?

L214 their size -> the magnitude

L218 So presumably the emission factors for different VOCs vary between the models? Please clarify.

Figure 2: are the maps emissions per 2.8 degree x 2.8 degree grid box?

Figure 3: Why show CH4 emissions?

L259 . . .when there parameters at. . . -> when these parameters are

L261 resulting -> results

Figure 4: Are the CO, NOx and VOC emissions really combined fire + biogenic + anthropogenic? Wouldn't it be clearer to just show how the fire emissions change, separately from other categories?

L285-290 The discussion of OH trends and NH/SH ratios is interesting, but seems a bit tangential? I suggest better integrate or remove.

L327 "The decrease in OH is the most likely reason for the simulated increase in CO and O3." This is a bit over-simplistic. BVOCs have increased. This will generate more CO and consume OH, as those extra VOCs are oxidised. Depending on the co-location of the VOCs, CO and NOx, this could either increase or decrease O3 – in this case it increases O3, indicating that the VOC and CO increases must be in areas with sufficient NOx to produce O3 (BVOC emissions in very low NOx regions can, at least

locally, decrease O3).

L340 "ice core observations" – I think these are oxygen isotope measurements from ice cores.

L344-345 0.4 +/- 0.2: the range here is a 5-95% confidence interval; 0.41 +/- 0.12: the range here is +/-1 standard deviation (i.e. encompassing 68% of the data). So these two are essentially the same, just using different range definitions. Please clarify this.
* * *

---

## Author Comment (AC1) · 20 Jun 2020

Response to reviewer comments for manuscript: **Tropospheric ozone radiative forcing uncertainty due to pre-industrial fire and biogenic emissions** by **Rowlinson et al.**

We thank the two reviewers for their detailed feedback on our manuscript. We have now carefully revised the manuscript according to all the comments provided. To guide the review process, we have copied the reviewer comments below (in black) and provided our responses (in blue).

**Responses to reviewer #1:**

**Reviewer Summary:**
In this study, the authors use a chemistry transport and different inventories of preindustrial fire and biogenic emissions to argue that the uncertainty range of ozone radiative forcing has been overestimated in past multi-model studies and assessments. The paper is the ozone counterpart to Hamilton et al. (2018), which made a similar point about biomass-burning aerosols.
The paper is very well written and structured in a straightforward way. The changes in simulated tropospheric ozone are well understood from differences in precursor emissions, so the question is whether the alternative sets of preindustrial emissions are a good guide to the overall uncertainty. This is where my concerns are, as detailed below. Addressing my comments may involve new simulations, so may represent major revisions.

**Authors' response:** We would like to thank the reviewer for the positive and constructive comments on our manuscript. We have now revised the manuscript to address the reviewer's concerns and added further information to clarify why certain decisions were made.

**Main Comments:**
1. My main concern with the study is that the PD/PI pairs used to estimate radiative forcing are not consistent. There is only one PD simulation, using the CMIP6 inventory. But shouldn't the SIMFIRE-BLAZE PI simulation be coupled with a SIMFIRE-BLAZE PD simulation? Shouldn't the LMfire PI simulation be coupled with an LMfire PD simulation? If the PD simulations differ from CMIP6 in the same way as the PI simulations, then the impact on radiative forcing would be small. I acknowledge that fire models (including those used to provide the CMIP6 inventory) are typically overfitted to present-day observations, so their PD simulations should share common patterns, but at least the PD and PI distributions would always be consistent in terms of the internal physics of the fire emissions.

**Authors' response:** Our experimental design with a single PD anchor point is driven by the research question addressed. The focus of our study is tropospheric ozone radiative forcing uncertainty due to PI fire and biogenic emissions. Changing the PD inventory adds an additional uncertainty from the PD dataset.

However, to address the reviewer's concern we have now performed a new model simulation, PD SIMFIRE-BLAZE, to explore the impact of the uncertainty in PD fire inventories on tropospheric ozone radiative forcing (RF). A PD simulation is not available for LMfire, a PI fire model not designed to undertake a PD simulation. We find that this additional uncertainty is very small. This agrees well with the fact that the PD tropospheric ozone (RE) has been shown to be well constrained by satellite observations (Rap et al., 2015), implying that the uncertainty in tropospheric ozone RF (i.e. PD RE - PI RE) caused by uncertainties in PD emission inventories is small.

We therefore now compare PD vs. PI simulations with both PD CMIP6 and PD SIMFIRE-BLAZE inventories. We find very similar PD tropospheric ozone burdens (31.0 DU for PD CMIP6 and 31.2 DU for PD SIMFIRE-BLAZE) and similar PI to PD RF when coupled to each PI inventory, see table below.

Comparison of $O_3$ RF from each PI emissions inventory relative to the two PD inventories, CMIP6 and SIMFIRE-BLAZE.

|  | Tropospheric $O_3$ RF (Wm$^{-2}$) |
|---|---|
| PD CMIP6 – PI CMIP6 | 0.38 |
| PD CMIP6 – PI SIMFIRE-BLAZE | 0.35 |
| PD CMIP6 – PI LMfire | 0.27 |
|  |  |
| PD SIMFIRE-BLAZE – PI CMIP6 | 0.38 |
| PD SIMFIRE-BLAZE – PI SIMFIRE-BLAZE | 0.36 |
| PD SIMFIRE-BLAZE – PI LMfire | 0.26 |

This is now discussed in the text and a comparison of the two PD simulations is included in Table 2 of the manuscript.

**Changes in manuscript:**
**L129-137**
"The PD simulations used anthropogenic emissions from the MACCity emissions dataset (from EU projects MACC/CityZEN; Lamarque et al. (2010)) and CCMI biogenic emissions (Sindelarova et al., 2014). Two PD simulations were performed, namely the primary PD simulation (PD CMIP) driven by the Global Fire Emissions Database version 4 with small fires (GFED v4s) inventory as employed in CMIP6 (Randerson et al., 2017; van Marle et al., 2017), and PD SIMFIRE-BLAZE (Knorr et al., 2014). A PD simulation is not available for LMfire, a PI fire model not designed to undertake a PD simulation. To isolate the effect of revised natural PI emissions on PI-to-PD tropospheric ozone RF, we compare the 6 PI simulations against the main PD CMIP6 simulation. The other PD simulation, i.e. PD SIMFIRE-BLAZE, was also included in our analysis in order to explore the additional uncertainty in RF introduced by PD emission inventories uncertainties. However, as PD tropospheric ozone RE was shown to be well constrained by satellite observation (Rap et al., 2015), this additional uncertainty is known to be small."

And clarified in the results section 3.4:

**Changes in manuscript:**
**L393-398**
"The estimated tropospheric $O_3$ RF, based on the CMIP6 PI and PD control simulations, is 0.38 Wm$^{-2}$ (Fig. 4 and Table 2), comparing well with the IPCC AR5 estimate of 0.4 ± 0.2 Wm$^{-2}$ (Myhre et al., 2013; Stevenson et al., 2013). We obtain the same 0.38 Wm$^{-2}$ RF value when contrasting the PI CMIP6 simulation against the other the other PD simulation (PD SIMFIRE-BLAZE). This is consistent with the fact that PD tropospheric $O_3$ is well constrained by satellite observations (Rap et al., 2015). Given the similarity of the PD simulations, the main PD CMIP6 simulation is used here as the PD for RF calculations in this section."

2. In a related concern, I note that section 2.6 implies that CCMI is a reasonable biogenic emission inventory for present-day because it compares well to flux measurements and other models. Then LPJ-GUESS is said to be similar to CCMI for present-day, implying it is also a reasonable inventory. Those are weak arguments, but there is at least an attempt at looking at performance of inventories. In contrast, section 2.4 on fire emission inventories does not discuss present-day performance. This is a problem because if SIMFIRE-BLAZE and/or LPJ-LMfire happen to be biased in an era where they can be constrained by observations, then the authors overstate the case for preindustrial emission uncertainty.

**Authors' response:** To address this concern, we have now conducted a comparison of PD emissions for the CMIP6 and SIMFIRE-BLAZE inventories and included this analysis in the manuscript (within the text, figures 1 and 2, and in Table 2). We believe this adds confidence in the reliability and relevance of the inventories.

**Changes in manuscript:**
**L164-171**
"The fire emissions in the PD SIMFIRE-BLAZE model are very similar to the PD CMIP6 inventory, with only slightly increased global NOx emissions (174 Tg/yr compared to 171 Tg/yr in CMIP6) and CO emissions (1027 Tg/yr compared to 970 Tg/yr). The global distribution of the inventories is also similar (Fig. 1), with slightly larger CO emissions in the SH tropics in PD SIMFIRE-BLAZE, but smaller in the NH tropical region. NOx and VOC emissions are similar in both inventories across all latitude bands (Fig. 1b, d). The seasonality of emissions is also consistent across both inventories in terms of NOx and VOC emissions, however for CO the peak in emissions is slightly later for the SIMFIRE-BLAZE inventory (Fig. 3). The slightly higher emissions in PD SIMFIRE-BLAZE result in a simulated tropospheric O3 burden of 360 Tg/yr, an increase of 1% relative to the PD CMIP6 TOMCAT-GLOMAP simulation (Table 2)."

**Other Comments:**
**Line 158**: "within the quantifiable uncertainty of fire emissions (Lee et al., 2013)". What do the authors mean here? For present-day or preindustrial? And is Lee et al. the correct reference? That paper does not mention LMFire at all.

**Authors' response:** Thank you for this comment as this point was not clear. The reference in question does not explicitly concern the LMfire inventory but finds substantial uncertainty in magnitude of emissions between inventories to be a common occurrence and estimates that uncertainty range for wildfire emissions is a factor of 4 larger/smaller. This is further supported by a recent study which found the total emission from 6 biomass burning datasets differed by a factor of 3.8 (Pan et al., 2020). This point is discussed with explicit reference to the relevant inventories in Hamilton et al. (2018), which should also have been included as a reference. This has now been corrected and the point reformulated more clearly in the text.

**Changes in manuscript:**
**L183-185**
"Although the PI LMfire and PI SIMFIRE-BLAZE emissions are substantially larger than the PI CMIP6 emissions, both inventories fall within the current uncertainty range for fire emissions, deemed to differ by up to a factor of ~4 (Lee et al., 2013; Hamilton et al., 2018; Pan et al., 2020)."

**Figure 1a**: LMfire has large CO emissions between 25 and 50S. What is burning there? Australia? Argentina?
**Authors' response:** As the reviewer correctly suggests, the increased CO emission between 25S and 50S is primarily due to increased burning in Australia in the LM fire emissions. Smaller increases in Argentina and South Africa also contribute to the relatively large change in emissions in LMfire, as shown in Figure 1.

[Figure]

**Figure 1. Global annual CO emissions in the (a) PD CMIP6 inventory, (b) PI CMIP6, (c) PI SIMFIRE-BLAZE and (d) PI LMfire. Red parallels indicate 20°S-50°S.**

**Changes in manuscript:**
**L228-229**
"The largest increase occurs due to increased SH burning in the LMfire inventory, substantially increasing CO emissions from Australia and South America (particularly Eastern Amazonia and Argentina)."

**Figures 2a, b, c**: What are those black lines in South America and Africa? In the difference maps, they seem to correspond to a brutal change in emissions, with differences between datasets switching sign suddenly.

**Authors' response:** We agree this was introducing some confusion - thank you for the comment. The black lines in question were actually topographical features (the Amazon and Congo Rivers), which are too prominent at that projection and resolution. The Figure 2 has now been updated so this is clear.

**Responses to reviewer #2:**

**General points**: This is an interesting study – and makes an important point: pre-industrial emissions from fires and biogenic sources are a major source of uncertainty for ozone radiative forcing. As explained below, it could benefit from some clarifications. In particular, why are these new estimates of PI emissions better than those used by CMIP6? Some details of the modelling need to be clarified – I was baffled by the discussion of CH4 emissions for simulations where I thought CH4 concentrations were prescribed. If the points below can be cleared up, then I am happy to recommend this paper should be accepted for publication in ACP.

**Authors' response:** We would like to thank the reviewer for their general comments on the manuscript and positive remarks on the study. We have endeavoured to address all specific comments and our responses and corrections are detailed below.

Specific comments:

L25 of up to -> by up to
**Authors' response:** Corrected.

L56 "human impact on. . . anthropogenic emissions. . ." Reword. I think we can be fairly sure there is a human impact on anthropogenic emissions. . .
**Authors' response:** This has been reworded to make the point more clearly.

**Changes in manuscript:**
**L53-56**
"While human activities such as deforestation, land-use change and fire management are known to affect natural emission sources of $O_3$ precursor gases, their impact on emissions net change remains very uncertain (Mickley et al., 2001; Arneth et al., 2010)."

L99 State thickness (metres or hPa) of the lowest model level.
**Authors' response:** We have now added this information within the text.

**Changes in manuscript:**
**L102**
"Biomass burning and biogenic emissions are emitted into the lowest model level, which extends from the surface to 951 hPa. "

L119 Do the prescribed surface CH4 concentrations have spatial variation, or just a constant value everywhere? Given later comments about CH4 emissions, please clarify further how CH4 is handled by the model.
**Authors' response:** We agree this should have been stated much more clearly. The global mean $CH_4$ concentration is scaled to observations for a particular year, but the spatial variation is maintained. Therefore, an emissions inventory is still required and spatial differences in $CH_4$ emissions between inventories are still relevant. We have now altered the text in the manuscript to make this clear.

**Changes in manuscript:**
**L99-101**
"The annual global mean surface $CH_4$ mixing ratio is scaled in TOMCAT-GLOMAP based on observed global surface mean concentration for the year being simulated; however, the spatial variation in $CH_4$ concentrations is maintained in the model."

L146 "Total PI fire emissions. . . SIMFIRE-BLAZE. . . are 28% larger than. . . PI CMIP6". It would be instructive to know PD fire emissions predicted by the SIMFIRE-BLAZE model. Can the model reproduce the present-day GFED distribution, or something similar? It may be that the higher PI values indicate a bias in this model towards higher values. It is hard to know how to verify or evaluate the PI fire emissions without some measure of the model's abilities – and presumably evaluation for present-day is the best evaluation possible. If this is not the case, then at least some discussion of how much faith we should have in these PI values is required.
L155 Similarly for the LPJ-LMfire model.

**Authors' response: T**his point is closely related to a comment from Reviewer 1. We have now conducted a comparison between PD emissions from CMIP6 and the SIMFIRE-BLAZE model, finding comparable emission magnitudes and distributions and resulting in very similar simulated tropospheric ozone concentrations (now included in Table 2 of the manuscript). We have also now quantified the effect of using the PD SIMFIRE-BLAZE emissions as the PD anchor for the RF calculations, finding similar RF as with the PD CMIP6 simulation (see table below).

Comparison of O3 RF from each PI emissions inventory relative to the two PD inventories, CMIP6 and SIMFIRE-BLAZE.

|  | Tropospheric $O_3$ RF (Wm$^{-2}$) |
|---|---|
| PD CMIP6 – PI CMIP6 | 0.38 |
| PD CMIP6 – PI SIMFIRE-BLAZE | 0.35 |
| PD CMIP6 – PI LMfire | 0.27 |
|  |  |
| PD SIMFIRE-BLAZE – PI CMIP6 | 0.38 |
| PD SIMFIRE-BLAZE – PI SIMFIRE-BLAZE | 0.36 |
| PD SIMFIRE-BLAZE – PI LMfire | 0.26 |

More detail on this has now been included in the manuscript.

**Changes in manuscript:**
**L129-137:**
"The PD simulations used anthropogenic emissions from the MACCity emissions dataset (from EU projects MACC/CityZEN; Lamarque et al., 2010) and CCMI biogenic emissions (Sindelarova et al., 2014). Two PD simulations were performed, namely the main PD simulation (PD CMIP6) driven by the Global Fire Emissions Database version 4s (GFEDv4s) inventory as employed in CMIP6 (Randerson et al., 2017; van Marle et al., 2017), and PD SIMFIRE-BLAZE which has been optimised against 3 global burned area datasets (Knorr et al., 2014). A PD simulation is not available for LMfire, a PI fire model not designed to undertake a PD simulation. To isolate the effect of revised natural PI emissions on PI-to-PD tropospheric ozone RF, we compare the 6 PI simulations against the main PD CMIP6 simulation. The other PD simulation, i.e. PD SIMFIRE-BLAZE, is also included in our analysis in order to explore the additional uncertainty in RF introduced by PD emission inventories uncertainties. However, as PD tropospheric ozone RE was shown to be well constrained by satellite observation (Rap et al., 2015), this additional uncertainty is known to be small."

**L163-170**

"The fire emissions in the PD SIMFIRE-BLAZE model are very similar to the PD CMIP6 inventory, with only slightly increased global NOx emissions (174 Tg yr-1 compared to 171 Tg yr-1 in CMIP6) and CO emissions (1027 Tg yr-1 compared to 970 Tg yr-1). The global distribution of the inventories is also similar (Fig. 1), with slightly larger CO emissions in the SH tropics in PD SIMFIRE-BLAZE, but smaller in the NH tropical region. NOx and VOC emissions are similar in both inventories across all latitude bands (Fig. 1b, d). The seasonality of emissions is also consistent across both inventories in terms of NOx and VOC emissions, however for CO there is a later and larger peak in emissions in the SIMFIRE-BLAZE inventory (Fig. 3). The small emission increases in PD SIMFIRE-BLAZE result in a simulated tropospheric O3 burden of 359.9 Tg, an increase of 1% relative to the PD CMIP6 TOMCAT-GLOMAP simulation (Table 2)."

Perhaps the key question here is whether the fire models used here are better than the fire models used in the CMIP6 base case. Are they clearly better, or are they just different? My non-expert reading of this is that they are just different. Please do try to convince me they are better.

**Authors' response:** We agree it is important to explain this better. The main purpose of our study is to quantify the impact of the existing large uncertainty in preindustrial natural emissions on tropospheric ozone radiative forcing. While there is not enough evidence to claim that one particular inventory outperforms all others in all regions, there is however evidence to suggest they are all plausible. Hamilton et al. (2018) made the case that the revised fire modelling inventories employed here arguably represent PI to PD changes in the paleoenvironmental archives of fire activity of the historical period with more accuracy than the CMIP6 inventory. Here, we add to the Hamilton et al. (2018) analysis by also comparing simulated CO from each inventory with ice-core records from the Wang et al. (2010) dataset. This comparison further supports the argument that the PI biomass burning emissions in CMIP6 are too small. We have now reformulated the text to better communicate this point, emphasising the improved performance in comparison to proxy records as clear indication that the revised inventories offer important insight to the uncertainties in tropospheric O$_3$ in the preindustrial atmosphere.

**Changes in manuscript:**
**L182-190**

"Although the PI LMfire and PI SIMFIRE-BLAZE emissions are substantially larger than the PI CMIP6 emissions, both inventories fall with the current uncertainty range for fire emissions, deemed to differ by up to a factor of ~4 (Lee et al., 2013; Hamilton et al., 2018; Pan et al., 2020). In Hamilton et al. (2018), both the SIMFIRE-BLAZE and LMfire PI inventories were shown to compare more favourably than CMIP6 to changes in PI to PD ice core BC measurements in the Swiss Alps. Furthermore, the LMfire emissions result in simulated aerosol concentrations that were closer to Northern Hemisphere (NH) ice core records in Greenland and Wyoming than both the CMIP6 and SIMFIRE-BLAZE emissions (Hamilton et al., 2018). In addition to the extensive examination of paleoenvironmental archives with PI fire emissions datasets by Hamilton et al. (2018), here we compared simulated annual mean surface PI CO concentrations in Antarctica for each fire emissions inventory using the Southern Hemisphere (SH) ice core CO record from Wang et al. (2010)."

**L199-203**

"The combined evaluation of these inventories in Hamilton et al. (2018) and here indicates that although the revised PI fire inventories differ considerably from each other and are substantially larger than CMIP6 in some regions, they result in simulated PI atmospheric concentrations that more closely represent the changes observed in paleoenvironmental archives of changes in Industrial Era fire activity than CMIP6 estimates do. Therefore, their respective impacts on PI tropospheric O3 concentrations and RF estimates need to be carefully considered."

Figure 1, and all the figures, are of a poor resolution. I can just about make out the necessary details, but these need to be improved for the final version.

**Authors' response:** The figures have been replaced with higher resolution images so the details should be clearer now.

In Figure 1c, the PD CMIP6 CH4 emissions from fire total 566.6 Tg. This sounds suspiciously high – isn't that more like the value for the total PD CH4 emission flux?

**Authors' response:** We thank the reviewer for identifying this error. The plot and value in question is indeed the PD emission of $CH_4$ from all PD sources, not just biomass burning. Emissions from all sources are used in the plot to demonstrate the shift in magnitude of emissions from PI to PD. This mistake has now been rectified:

**Changes in manuscript:**
**Figure 1 caption:**
"Annual latitudinal mean preindustrial emissions (in Tg/yr) of (a) CO, (b) NOx, (c) CH4 and (d) VOCs), in PD CMIP6 (solid black line), PD SIMFIRE-BLAZE (dashed black), PI CMIP6 (dashed green), PI SIMFIRE-BLAZE (dotted orange), PI LMfire (dashed purple) inventories,. In (e), annual latitudinal mean BVOC emissions in (Tg/yr) in PD CCMI (solid black line), PD LPJ-GUESS (dashed dark green), PI LPJ-GUESS (dotted light green)."

**L220-222**
"Figure 1a-d shows annual latitudinal emissions of CO, NOx, CH4 and VOCs from all sources for the different fire inventories considered, while Figure 1e compares BVOC emissions (i.e. isoprene and monoterpenes) from the biogenic emissions inventories."

L192 delete PI.
**Authors' response:** Corrected.

L193 "The main driver of this increase [in fire emissions] is industrial emissions. . ." This must be wrong?
**Authors' response:** As clarified above this does refer to the PI to PD change in emissions from all sources, where the most important driver is in fact anthropogenic emissions from industry. This is now made clear with the updated plot caption and in the text.

L210 I don't understand why CH4 emissions are presented and discussed; surely if CH4 concentrations are prescribed, the CH4 emissions are irrelevant and redundant? Am I missing something?
**Authors' response:** We agree this should be clarified to avoid confusions. As mentioned in the response to an earlier comment, while the global mean CH4 concentration is scaled, the spatial variation is maintained. Therefore, simulated CH4 will vary spatially between simulations with different CH4 emissions. However, we acknowledge that due to the scaling the impact of changes to CH4 emissions on ozone formation is likely to be small. We have made this clear now in the text.

**Changes in manuscript:**
**L99-101**
"The global mean surface CH4 mixing ratio is scaled in TOMCAT-GLOMAP to a best estimate based on observed global surface mean concentration for the year being simulated, meaning that spatial variation in CH4 concentrations is maintained."

**L244-245**
"Due to the scaling of global mean surface CH4 concentrations in TOMCAT-GLOMAP, the
effect of changes in amount of CH4 emitted is likely small, however the change in
distribution may impact the formation and loss rates of tropospheric O3."

L214 their size -> the magnitude
**Authors' response:** Corrected.

L218 So presumably the emission factors for different VOCs vary between the models?
Please clarify.
**Authors' response:** As the reviewer suggests, VOC emission factors do vary between
models, although differences in burned area and vegetation type also contribute to the
differences in VOC emission. This is now clarified in the text.

**Changes in manuscript:**
**L248-254**
"In terms of fire-emitted VOC species, their magnitude and distribution of emissions are fairly
consistent between PD and PI inventories. PI CMIP6 are 87% of PD CMIP6 values, with PI
SIMFIRE-BLAZE at 97% (303 Tg/yr). Total global VOC emissions are largest in LMfire at
349 Tg/yr, 29% larger than PI CMIP6 (271 Tg/yr) and 13% larger than PD CMIP6 (310
Tg/yr). The distribution of total global VOC emissions is relatively uniform across all
inventories; however individual species do have larger variability between inventories.
Formaldehyde and acetylene for example have substantially increased SH emissions in
SIMFIRE-BLAZE and LMfire, due to differences in emission factors, vegetation type and
burned area between the fire models."

Figure 2: are the maps emissions per 2.8 degree x 2.8 degree grid box?
**Authors' response:** The figure shows the emissions on a 1°×1° resolution. The emissions
are regridded to the TOMCAT resolution of 2.8 x 2.8 within the model. This is now made
clear in the figure caption.

**Changes in manuscript:**
**Figure 2 caption**
"Annual BVOC (isoprene + monoterpenes) emissions at 1°×1° in the two present-day
biogenic emissions inventories (CCMI and LPJ-GUESS) and the preindustrial LPJ-GUESS
inventory. Top panels (a-c) show total emissions per year, while lower panels (d-f) show
differences between the three inventories. Total annual emissions and difference in annual
emissions are also shown."

Figure 3: Why show CH4 emissions?
**Authors' response:** This point is addressed in an earlier response.

L259 . . .When there parameters at. . . -> when these parameters are
**Authors' response:** Corrected.

L261 resulting -> results
**Authors' response:** Corrected.

Figure 4: Are the CO, NOx and VOC emissions really combined fire + biogenic +
anthropogenic? Wouldn't it be clearer to just show how the fire emissions change, separately
from other categories?

**Authors' response:** Yes, the emissions magnitudes in the figure are the combined totals.
We agree there are different ways one could present our results in this figure and we have

considered a few options. In the end we decided on this version as it illustrates how each sector contributes to tropospheric $O_3$ formation, as well as displaying the differences between simulations and results. We feel this figure contributes to the study by adding a lot of information in a manner that is easy to interpret, for various levels of expertise.

L285-290 The discussion of OH trends and NH/SH ratios is interesting, but seems a bit tangential? I suggest better integrate or remove.

**Authors' response:** We agree this discussion was indeed a bit tangential and did not add substantially to the manuscript other than to confirm the relatively large SH emission increase in the LMfires inventory. We followed the reviewer's suggestion and have now removed it from the revised manuscript. The rest of the discussion of OH changes has also been shortened and rewritten to make the relevance of the discussion clearer.

**Changes in manuscript:**
**L320-328**
 "The hydroxyl radical (OH), which plays a key role in regulating tropospheric O3 concentrations, had lower PI concentrations than in PD due to the higher concentrations of OH precursors $NO_x$ and $O_3$ in PD outcompeting the effect of increased $CH_4$ and CO concentrations which deplete OH (Naik et al., 2013). This is consistent in the TOMCAT PI simulations, with airmass-weighted global mean concentrations of tropospheric OH, at 1.06, 1.06 and 1.11 $\times 10^6$ molecules cm$^{-3}$ in CMIP6, SIMFIRE-BLAZE and LMfire, respectively, compared to 1.12 $\times 10^6$ molecules cm$^{-3}$) in PD CMIP6. Each of these values fall within one standard deviation of the Atmospheric Chemistry and Climate Model Intercomparison Project (ACCMIP) multi-model mean of 1.13 ± 0.17 (Naik et al., 2013)."

L327 "The decrease in OH is the most likely reason for the simulated increase in CO and O3."
This is a bit over-simplistic. BVOCs have increased. This will generate more CO and consume OH, as those extra VOCs are oxidised. Depending on the colocation of the VOCs, CO and NOx, this could either increase or decrease O3 – in this case it increases O3, indicating that the VOC and CO increases must be in areas with sufficient NOx to produce O3 (BVOC emissions in very low NOx regions can, at least locally, decrease O3).

**Authors' response:** Thank you for pointing this out - we agree this needs to be better explained. We have now expanded to include additional detail.

**Changes in manuscript:**
**369-372**
"The decrease in OH is the likely responsible for the simulated increase in CO, as OH is consumed by VOC oxidation. The increase in global tropospheric $O_3$ indicates that the simulated increases in VOC and CO concentrations are co-located with high $NO_x$ concentrations, as in low $NO_x$ BVOCs may decrease local $O_3$ concentrations."

L340 "ice core observations" – I think these are oxygen isotope measurements from ice cores.
**Authors' response:** Corrected.

L344-345 0.4 +/- 0.2: the range here is a 5-95% confidence interval; 0.41 +/- 0.12: the range here is +/-1 standard deviation (i.e. encompassing 68% of the data). So these two are essentially the same, just using different range definitions. Please clarify this.
**Authors' response:** Thank you for pointing out this error. We have now simplified this sentence in the manuscript.

**Changes in manuscript:**
**L387-388**

[revised manuscript text omitted]

---

## Author Comment (AC2) · 20 Jun 2020

**Tropospheric ozone radiative forcing uncertainty due to pre-industrial fire and biogenic emissions**

M. J. Rowlinson[1,2], A. Rap[1], D. S. Hamilton[3], R. J. Pope[1,2], S. Hantson[4,5], S. R. Arnold[1], J. O. Kaplan[6], A. Arneth[4], M. P. Chipperfield[1,2], P. M. Forster[7], L. 
[revised manuscript text omitted]